# Exact Recovery of Hard Thresholding Pursuit

**Xiao-Tong Yuan**
B-DAT Lab
Nanjing University of Info. Sci.&Tech.
Nanjing, Jiangsu, 210044, China
xtyuan@nuist.edu.cn

**Ping Li**[†‡]    **Tong Zhang**[†]
†Depart. of Statistics and ‡Depart. of CS
Rutgers University
Piscataway, NJ, 08854, USA
{pingli,tzhang}@stat.rutgers.edu

## Abstract

The Hard Thresholding Pursuit (HTP) is a class of truncated gradient descent methods for finding sparse solutions of $\ell_0$-constrained loss minimization problems. The HTP-style methods have been shown to have strong approximation guarantee and impressive numerical performance in high dimensional statistical learning applications. However, the current theoretical treatment of these methods has traditionally been restricted to the analysis of parameter estimation consistency. It remains an open problem to analyze the support recovery performance (a.k.a., sparsistency) of this type of methods for recovering the global minimizer of the original NP-hard problem. In this paper, we bridge this gap by showing, for the first time, that exact recovery of the global sparse minimizer is possible for HTP-style methods under restricted strong condition number bounding conditions. We further show that HTP-style methods are able to recover the support of certain relaxed sparse solutions without assuming bounded restricted strong condition number. Numerical results on simulated data confirms our theoretical predictions.

## 1 Introduction

In modern high dimensional data analysis tasks, a routinely faced challenge is that the number of collected samples is substantially smaller than the dimensionality of features. In order to achieve consistent estimation in such small-sample-large-feature settings, additional assumptions need to be imposed on the model. Among others, the low-dimensional structure prior is the most popular assumption made in high dimensional analysis. This structure can often be captured by imposing sparsity constraint on model space, leading to the following $\ell_0$-constrained minimization problem:

$$\min_{x \in \mathbb{R}^p} f(x), \quad \text{s.t. } \|x\|_0 \leq k, \tag{1}$$

where $f : \mathbb{R}^p \mapsto \mathbb{R}$ is a smooth convex loss function and $\|x\|_0$ denotes the number of nonzero entries in $x$. Due to the cardinality constraint, Problem (1) is not only non-convex, but also NP-hard in general (Natarajan, 1995). Thus, it is desirable to develop efficient computational procedures to approximately solve this problem.

When the loss function is squared regression error, Problem (1) reduces to the compressive sensing problem (Donoho, 2006) for which a vast body of greedy selection algorithms have been proposed including orthogonal matching pursuit (OMP) (Pati et al., 1993), compressed sampling matching pursuit (CoSaMP) (Needell & Tropp, 2009), hard thresholding pursuit (HTP) (Foucart, 2011) and iterative hard thresholding (IHT) (Blumensath & Davies, 2009) to name a few. The greedy algorithms designed for compressive sensing can usually be generalized to minimize non-quadratic loss functions (Shalev-Shwartz et al., 2010; Yuan & Yan, 2013; Bahmani et al., 2013). Comparing to those convex-relaxation-based methods (Beck & Teboulle, 2009; Agarwal et al., 2010), these greedy se-

lection algorithms often exhibit similar accuracy guarantees but more attractive computational efficiency and scalability.

Recently, the HTP/IHT-style methods have gained significant interests and they have been witnessed to offer the fastest and most scalable solutions in many cases (Yuan et al., 2014; Jain et al., 2014). The main theme of this class of methods is to iteratively perform gradient descent followed by a truncation operation to preserve the most significant entries, and an (optional) debiasing operation to minimize the loss over the selected entries. In (Blumensath, 2013; Yuan et al., 2014), the rate of convergence and parameter estimation error of HTP/IHT-style methods were established under proper Restricted Isometry Property (RIP) (or restricted strong condition number) bound conditions. Jain et al. (2014) presented and analyzed several relaxed variants of HTP/IHT-style algorithms for which the estimation consistency can be established without requiring the RIP conditions. Very recently, the extensions of HTP/IHT-style methods to structured and stochastic sparse learning problems have been investigated in (Jain et al., 2016; Li et al., 2016; Shen & Li, 2016).

## 1.1 An open problem: exact recovery of HTP

In this paper, we are particularly interested in the exact recovery and support recovery performance of the HTP-style methods. A pseudo-code of HTP is outlined in Algorithm 1 which is also known as GraHTP in (Yuan et al., 2014). Although this type of methods have been extensively analyzed in the original paper (Foucart, 2011) for compressive sensing and several recent followup work (Yuan et al., 2014; Jain et al., 2014, 2016) for generic sparse minimization, the state-of-the-art is only able to derive convergence rates and parameter estimation error bounds for HTP. It remains an open and challenging problem to analyze its ability to exactly recover the global sparse minimizer of Problem (1) in general settings. Actually, the support/structure recovery analysis is the main challenge in many important sparsity models including compressive sensing and graphical models learning (Jalali et al., 2011; Ravikumar et al., 2011): once the support is recovered, computing the actual nonzero coefficients just boils down to solving a convex minimization problem restricted on the supporting set.

Since the output of HTP is always $k$-sparse, the existing estimation error results in (Foucart, 2011; Yuan et al., 2014; Jain et al., 2014) naturally imply some support recovery conditions. For example, for perfect measurements, the results in (Foucart, 2011; Yuan et al., 2014) guarantee that HTP can exactly recover the underlying true sparse model parameters. For noisy models, roughly speaking, as long as the smallest (in magnitude) nonzero entry of the $k$-sparse minimizer of (1) is larger than the estimation error bound of HTP, an exact recovery of the minimizer can be guaranteed. However, these pieces of support recovery results implied by the estimation error bound turn out to be loose when compared to the main results we will derive in the current paper.

---

**Algorithm 1:** Hard Thresholding Pursuit.

**Input** : Loss function $f(x)$, sparsity level $k$, step-size $\eta$.
**Initialization** $x^{(0)} = 0$, $t = 1$.
**Output** : $x^{(t)}$.
**repeat**
    (S1) Compute $\tilde{x}^{(t)} = x^{(t-1)} - \eta \nabla f(x^{(t-1)})$;
    (S2) Select $F^{(t)} = \text{supp}(\tilde{x}^{(t)}, k)$ be the indices of $\tilde{x}^{(t)}$ with the largest $k$ absolute values;
    (S3) Compute $x^{(t)} = \arg\min\{f(x), \text{supp}(x) \subseteq F^{(t)}\}$;
    (S4) Update $t \leftarrow t + 1$;
**until** $F^{(t)} = F^{(t-1)}$;

---

## 1.2 Overview of our results

The core contribution in this work is a deterministic support recovery analysis of HTP-style methods which to our knowledge has not been systematically conducted elsewhere in literature. Our *first* result (see Theorem 1) shows that HTP as described in Algorithm 1 is able to exactly recover the $k$-sparse minimizer $x^\star = \arg\min_{\|x\|_0 \leq k} f(x)$ if $x^\star_{\min}$, i.e., the smallest non-zero entry of $x^\star$, is significantly larger than $\|\nabla f(x^\star)\|_\infty$ and certain RIP-type condition can be fulfilled as well. Moreover, the exact recovery can be guaranteed in finite running of Algorithm 1 with geometric rate of convergence. Our *second* result (see Theorem 2) shows that the support recovery of an arbitrary $k$-sparse

Table 1: Comparison between our results and several prior results on HTP-style algorithms.

| Related Work | Target Solution | RIP Condition Free | Support Recovery |
|---|---|---|---|
| (Foucart, 2011) | True $k$-sparse signal $x$ | $\times$ | $\times$ |
| (Yuan et al., 2014) | Arbitrary $\bar{x}$ with $\|\bar{x}\|_0 \leq k$ | $\times$ | $\times$ |
| (Jain et al., 2014) | $\bar{x} = \arg\min_{\|x\|_0 \leq \bar{k}} f(x)$ for proper $\bar{k} \ll k$ | $\sqrt{}$ | $\times$ |
| **Ours** | Arbitrary $\bar{x}$ with $\|\bar{x}\|_0 \leq k$ | $\times$ (for $\|\bar{x}\|_0 = k$), $\sqrt{}$ (for $\|\bar{x}\|_0 \ll k$) | $\sqrt{}$ |

vector $\bar{x}$ can be guaranteed if $\bar{x}_{\min}$ is well discriminated from $\sqrt{k}\|\nabla f(x^\star)\|_\infty$ or $\|\nabla f(x^\star)\|_\infty$, pending on the optimality of $\bar{x}$ over its own supporting set. Our *third* result (see Theorem 3) shows that HTP is able to recover the support of certain relaxed sparse minimizer $\bar{x}$ with $\|\bar{x}\|_0 \ll k$ under an arbitrary restricted strong condition number. More formally, given the restricted strong smoothness/convexity (see Definition 1) constants $M_{2k}$ and $m_{2k}$, the recovery of supp$(\bar{x})$ is possible if $k \geq (1 + 16M_{2k}^2/m_{2k}^2)\bar{k}$ and the smallest non-zero element in $\bar{x}$ is significantly larger than the rooted objective value gap $\sqrt{f(\bar{x}) - f(x^\star)}$. The support recovery can also be guaranteed in finite iteration for this case. By specifying our deterministic analysis to least squared regression and logistic regression, we are able to obtain the sparsistency guarantees of HTP for these statistical learning examples. Monte-Carlo simulation results confirm our theoretical predictions. Table 1 summarizes a high-level comparison between our work and the state-of-the-art analysis for HTP-style methods.

### 1.3 Notation and organization

**Notation** Let $x \in \mathbb{R}^p$ be a vector and $F$ be an index set. We denote $[x]_i$ the $i$th entry of vector $x$, $x_F$ the restriction of $x$ to index set $F$ and $x_k$ the restriction of $x$ to the top $k$ (in absolute vale) entries. The notation supp$(x)$ represents the index set of nonzero entries of $x$ and supp$(x, k)$ represents the index set of the top $k$ (in absolute vale) entries of $x$. We conventionally define $\|x\|_\infty = \max_i |[x]_i|$ and define $x_{\min} = \min_{i \in \text{supp}(x)} |[x]_i|$.

**Organization** This paper proceeds as follows: In §2, we analyze the exact recovery performance of HTP. The applications of our analysis to least squared regression and logistic regression models are presented in §3. Monte-Carlo simulation results are reported in §4. We conclude this paper in §5. Due to space limit, all the technical proofs of our results are deferred to an appendix section which is included in the supplementary material.

## 2 A Deterministic Exact Recovery Analysis

In this section, we analyze the exact support recovery performance of HTP as outlined in Algorithm 1. In large picture, the theory developed in this section can be decomposed into the following three ingredients:

- First, we will investigate the support recovery behavior of the global $k$-sparse minimizer $x^\star = \arg\min_{\|x\|_0 \leq k} f(x)$. The related result is summarized in Proposition 1.

- Second, we will present in Theorem 1 the guarantee of HTP for exactly recovering $x^\star$.

- Finally, by combining the the above two results we will be able to establish the support recovery result of HTP in Theorem 2. Furthermore, we derive an RIP-condition-free support recovery result in Theorem 3.

Our analysis relies on the conditions of Restricted Strong Convexity/Smoothness (RSC/RSS) which are conventionally used in previous analysis for HTP (Yuan et al., 2014; Jain et al., 2014).

**Definition 1** (Restricted Strong Convexity/Smoothness). *For any integer $s > 0$, we say $f(x)$ is restricted $m_s$-strongly convex and $M_s$-smooth if there exist $m_s, M_s > 0$ such that*

$$\frac{m_s}{2}\|x - y\|^2 \leq f(x) - f(y) - \langle \nabla f(y), x - y \rangle \leq \frac{M_s}{2}\|x - y\|^2, \quad \forall \|x - y\|_0 \leq s. \quad (2)$$

The ratio $M_s/m_s$, which measures the curvature of the loss function over sparse subspaces, will be referred to as *restricted strong condition number* in this paper.

## 2.1 Preliminary: Support recovery of $x^\star$

Given a target solution $\bar{x}$, the following result establishes some sufficient conditions under which $x^\star$ is able to exactly recover the supporting set of $\bar{x}$. A proof of this result is provided in Appendix B (see the supplementary file).

**Proposition 1.** *Assume that $f$ is $M_{2k}$-smooth and $m_{2k}$-strongly convex. Let $\bar{x}$ be an arbitrary $k$-sparse vector. Let $\bar{x}^\star = \arg\min_{supp(x) \subseteq supp(\bar{x})} f(x)$ and $\bar{l} > 0$ be a scalar such that*

$$f(\bar{x}^\star) = f(\bar{x}) + \langle \nabla f(\bar{x}), \bar{x}^\star - \bar{x} \rangle + \frac{\bar{l}}{2} \|\bar{x}^\star - \bar{x}\|_1^2.$$

*Then we have $supp(\bar{x}) = supp(x^\star)$ if either of the following two conditions is satisfied:*

*(1) $\bar{x}_{\min} \geq \frac{2\sqrt{2k}}{m_{2k}} \|\nabla f(\bar{x})\|_\infty$;*

*(2) $\bar{x}_{\min} \geq \left( \frac{\bar{\vartheta}}{M_{2k}} + \frac{2\bar{\vartheta}+2}{\bar{l}} \right) \|\nabla f(\bar{x})\|_\infty$, $\frac{m_{2k}}{M_{2k}} \geq \max\left\{ \frac{3\bar{\vartheta}+1}{4\bar{\vartheta}}, \frac{\sqrt{3}}{2} \right\}$, for some $\bar{\theta} > 1$.*

**Remark 1.** *The quantity $\bar{l}$ actually measures the strong-convexity of $f$ at the point $(\bar{x}^\star - \bar{x})$ in $\ell_1$-norm. From its definition we can verify that $\bar{l}$ is valued in the interval $[m_{2k}/k, M_{2k}]$ if $\bar{x} \neq \bar{x}^\star$. The closer $\bar{l}$ is to $M_{2k}$, the weaker lower bound condition can be imposed on $\bar{x}_{\min}$ in the condition (2). In (Nutini et al, 2015), a similar strong-convexity measurement has been defined over the entire vector space for refined convergence analysis of the coordinate descent methods. Different from (Nutini et al, 2015), we only require such an $\ell_1$-norm strong-convexity condition holds at certain target points of interest. Particularly if $\bar{x} = \bar{x}^\star$, i.e., $\bar{x}$ is optimal over its supporting set, then we may simply set $\bar{l} = \infty$ in Proposition 1.*

## 2.2 Main results: Support recovery of HTP

Equipped with Proposition 1, it will be straightforward to guarantee the support recovery of HTP if we can derive sufficient conditions under which HTP is able to exactly recover $x^\star$. Denote $F^\star = supp(x^\star)$. Intuitively, $x^\star_{\min}$ should be significantly larger than $\|\nabla f(x^\star)\|_\infty$ to attract HTP to be stuck at $x^\star$ (see Lemma 5 in Appendix B for a formal elaboration). The exact recovery analysis also relies on the following quantity $\triangle^{-\star}$ which measures the gap between the minimal $k$-sparse objective value $f(x^*)$ and the remaining ones over supporting sets other than $supp(x^*)$:

$$\triangle^{-\star} := f(x^{-\star}) - f(x^\star),$$

where $x^{-\star} = \arg\min_{\|x\|_0 \leq k, supp(x) \neq supp(x^\star), f(x) > f(x^\star)} f(x)$. Intuitively, the larger $\triangle^{-\star}$ is, the easier and faster $x^\star$ can be recovered by HTP. It is also reasonable to expect that the step-size $\eta$ should be well bounded away from zero to avoid undesirable early stopping.

Inspired by these intuitive points, we present the following theorem which guarantees the exact recovery of HTP when the restricted strong condition number is well bounded. A proof of this theorem is provided in Appendix C (see the supplementary file).

**Theorem 1.** *Assume that $f$ is $M_{2k}$-smooth and $m_{2k}$-strongly convex. Assume that $\vartheta^\star := \frac{M_{2k} x^\star_{\min}}{\|\nabla f(x^\star)\|_\infty} > 1$ and $\frac{m_{2k}}{M_{2k}} \geq \frac{7\vartheta^\star + 1}{8\vartheta^\star}$. If we set the step-size to be $\eta = \frac{m_{2k}}{M_{2k}^2}$, then the optimal $k$-sparse solution $x^\star$ is unique and HTP will terminate with output $x^{(t)} = x^\star$ after at most*

$$t = \left\lceil \frac{M_{2k}^3}{m_{2k}^2 (M_{2k} - m_{2k})} \ln \frac{\triangle^{(0)}}{\triangle^{-\star}} \right\rceil$$

*steps of iteration, where $\triangle^{(0)} = f(x^{(0)}) - f(x^\star)$ and $\triangle^{-\star} = \min_{\|x\|_0 \leq k, supp(x) \neq supp(x^\star), f(x) > f(x^\star)} \{f(x) - f(x^\star)\}$.*

**Remark 2.** *Theorem 1 suggests that HTP is able to exactly recover $x^\star$ provided that $x^\star_{\min}$ is strictly larger than $\|\nabla f(x^\star)\|_\infty / M_{2k}$ and the restricted strong condition number is well bounded, i.e., $M_{2k}/m_{2k} \leq \frac{8\theta^\star}{7\theta^\star + 1} < 1.14$.*

As a consequence of Proposition 1 and Theorem 1, the following theorem establishes the performance of HTP for recovering the support of an arbitrary $k$-sparse vector. A proof of this result is provided in Appendix D (see the supplementary file).

**Theorem 2.** *Let $\bar{x}$ be an arbitrary $k$-sparse vector and $\bar{l}$ be defined in Proposition 1. Assume that the conditions in Theorem 1 hold. Then HTP will output $x^{(t)}$ satisfying $supp(x^{(t)}) = supp(\bar{x})$ in finite iteration, provided that either of the following two conditions is satisfied in addition:*

$$(1)\ \bar{x}_{\min} \geq \frac{2\sqrt{2k}}{m_{2k}}\|\nabla f(\bar{x})\|_{\infty}; \quad (2)\ \bar{x}_{\min} \geq \left(\frac{\vartheta^{\star}}{M_{2k}} + \frac{2\vartheta^{\star}+2}{\bar{l}}\right)\|\nabla f(\bar{x})\|_{\infty}.$$

In the following theorem, we further show that for proper $\bar{k} < k$, HTP method is able to recover the support of certain desired $\bar{k}$-sparse vector without assuming bounded restricted strong condition numbers. A proof of this theorem can be found in Appendix E (see the supplementary file).

**Theorem 3.** *Assume that $f$ is $M_{2k}$-smooth and $m_{2k}$-strongly convex. Let $\bar{x}$ be an arbitrary $\bar{k}$-sparse vector satisfying $k \geq \left(1 + \frac{16M_{2k}^2}{m_{2k}^2}\right)\bar{k}$. Set the step-size to be $\eta = \frac{1}{2M_{2k}}$.*

*(a) If $\bar{x}_{\min} > \sqrt{\frac{2(f(\bar{x})-f(x^{\star}))}{m_{2k}}}$, then HTP will terminate in finite iteration with output $x^{(t)}$ satisfying $supp(\bar{x}) \subseteq supp(x^{(t)})$.*

*(b) Furthermore, if $\bar{x}_{\min} > 1.62\sqrt{\frac{2(f(\bar{x})-f(x^{\star}))}{m_{2k}}}$, then HTP will terminate in finite iteration with output $x^{(t)}$ satisfying $supp(x^{(t)}, \bar{k}) = supp(\bar{x})$.*

**Remark 3.** *The main message conveyed by the part (a) of Theorem 3 is: If the nonzero elements in $\bar{x}$ are significantly larger than the rooted objective value gap $\sqrt{f(\bar{x})-f(x^{\star})}$, then $supp(\bar{x}) \subseteq supp(x^{(t)})$ can be guaranteed by HTP with sufficiently large sparsity level $k$. Intuitively, the closer $f(\bar{x})$ is to $f(x^{\star})$, the easier the conditions can be satisfied. Given that $f(\bar{x})$ is close enough to the unconstrained global minimizer of $f$ (i.e., the global minimizer of $f$ is nearly sparse), we will have $f(\bar{x})$ close enough to $f(x^{\star})$ since $f(\bar{x}) - f(x^{\star}) \leq f(\bar{x}) - \min_x f(x)$. In the ideal case where the sparse vector $\bar{x}$ is an unconstrained minimum of $f$, we will have $f(\bar{x}) = f(x^{\star})$, and thus $supp(\bar{x}) \subseteq supp(x^{(t)})$ holds under arbitrarily large restricted strong condition number.*

*The part (b) of Theorem 3 shows that under almost identical conditions (up to a slightly increased numerical constant) to those in Part(a), HTP will output $x^{(t)}$ of which the top $\bar{k}$ entries are exactly the supporting set of $\bar{x}$. The implication of this result is: in order to recover certain $\bar{k}$-sparse signals, one may run HTP with a properly relaxed sparsity level $k$ until convergence and then preserve the top $\bar{k}$ entries of the $k$-sparse output as the final $\bar{k}$-sparse solution.*

## 2.3 Comparison against prior results

It is interesting to compare our support recovery results with those implied by the parameter estimation error bounds obtained in prior work (Yuan et al., 2014; Jain et al., 2014). Actually, parameter estimation error bound naturally leads to the so called *x-min* condition which is key to the support recovery analysis. For example, it can be derived from the bounds in (Yuan et al., 2014) that under proper RIP condition $\|x^{(t)} - \bar{x}\| = \mathcal{O}(\sqrt{k}\|\nabla f(\bar{x})\|_{\infty})$ when $t$ is sufficiently large. This implies that as long as the $\bar{x}_{\min}$ is significantly larger than such an estimation error bound, exact recovery of $\bar{x}$ can be guaranteed. In the meantime, the results in (Jain et al., 2014) show that for some $\bar{k}$-sparse minimizer of (1) with $\bar{k} = \mathcal{O}\left(\frac{m_{2k}^2}{M_{2k}^2}k\right)$, it holds for arbitrary restrictive strong condition number that $\|x^{(t)} - \bar{x}\| = \mathcal{O}(\sqrt{k}\|\nabla f(\bar{x})\|_{\infty})$ when $t$ is sufficiently large. Provided that $\bar{x}_{\min}$ is significantly larger than such an error bound, it will hold true that $supp(\bar{x}) \subseteq supp(x^{(t)})$. Table 2 summarizes our support recovery results and those implied by the state-of-the-art results regarding target solution, dependency on RIP-type conditions and x-min condition. From this table, we can see that the x-min condition in Theorem 1 for recovering the global minimizer $x^{\star}$ is weaker than those implied in (Yuan et al., 2014) in the sense that the former is not dependent on a factor $\sqrt{k}$. Also our x-min condition in Theorem 3 is weaker than those implied in (Jain et al., 2014) because; 1) our bound $\mathcal{O}(\sqrt{f(\bar{x})-f(x^{\star})})$ is not explicitly dependent on a multiplier $\sqrt{k}$; and 2) it can be verified from the restricted strong-convexity of $f$ that $\sqrt{f(\bar{x})-f(x^{\star})} \leq \sqrt{k}\|\nabla f(\bar{x})\|_{\infty}/\sqrt{2m_{2k}}$.

Table 2: Comparison between our support recovery conditions and those implied by the existing estimation error bounds for HTP-style methods.

| Results | Target Solution | RIP Cond. | X-min Condition |
|---------|-----------------|-----------|-----------------|
| (Yuan et al., 2014) | Arbitrary $k$-sparse $\bar{x}$ | Required | $\bar{x}_{\min} > \mathcal{O}(\sqrt{k}\|\nabla f(\bar{x})\|_\infty)$ |
| (Jain et al., 2014) | $\|\bar{x}\|_0 = \mathcal{O}\left((\frac{m_{2k}}{M_{2k}})^2 k\right)$ | Free | $\bar{x}_{\min} > \mathcal{O}(\sqrt{k}\|\nabla f(\bar{x})\|_\infty)$ |
| **Theorem 1** | $x^\star = \arg\min_{\|x\|_0 \le k} f(x)$ | Required | $x^\star_{\min} > \mathcal{O}(\|\nabla f(x^\star)\|_\infty)$ |
| **Theorem 2** | Arbitrary $k$-sparse $\bar{x}$ | Required | $\bar{x}_{\min} > \mathcal{O}(\sqrt{k}\|\nabla f(\bar{x})\|_\infty)$ or $\bar{x}_{\min} > \mathcal{O}(\|\nabla f(\bar{x})\|_\infty)$ |
| **Theorem 3** | $\|\bar{x}\|_0 = \mathcal{O}\left((\frac{m_{2k}}{M_{2k}})^2 k\right)$ | Free | $\bar{x}_{\min} > \mathcal{O}\left(\sqrt{f(\bar{x}) - f(x^\star)}\right)$ |

It is also interesting to compare the support recovery result in Proposition 1 with those known for the following $\ell_1$-regularized estimator:

$$\min_{x \in \mathbb{R}^p} f(x) + \lambda\|x\|_1,$$

where $\lambda$ is the regularization strength parameter. Recently, a unified sparsistency analysis for this type of convex-relaxed estimator was provided in (Li et al., 2015). We summarize in below a comparison between our Proposition 1 and the state-of-the-art results in (Li et al., 2015) with respect to several key conditions:

- Local structured smoothness/convexity condition: Our analysis only requires first-order local structured smoothness/convexity conditions (i.e., RSC/RSS) while the analysis in (Li et al., 2015, Theorem 5.1, Condition 1) relies on certain second-order and third-order local structured smoothness conditions.

- Irrepresentablility condition: Our analysis is free of the Irrepresentablility Condition which is usually required to guarantee the sparsistency of $\ell_1$-regularized estimators (Li et al., 2015, Theorem 5.1, Condition 3).

- RIP-type condition: The analysis in (Li et al., 2015) is free of RIP-type condition, while ours is partially relying on such a condition (see Condition (2) of Proposition 1).

- X-min condition: Comparing to the x-min condition required in (Li et al., 2015, Theorem 5.1, Condition 4), which is of order $\mathcal{O}(\sqrt{k}\|\nabla f(\bar{x})\|_\infty)$, the x-min condition (1) in Proposition 1 is at the same order while the x-min condition (2) is weaker as it is not explicitly dependent on $\sqrt{k}$.

## 3 Applications to Statistical Learning Models

In this section, we apply our support recovery analysis to several sparse statistical learning models, deriving concrete sparsistency conditions in each case. Given a set of $n$ independently drawn data samples $\{(u^{(i)}, v^{(i)})\}_{i=1}^n$, we are interested in the following sparsity-constrained empirical loss minimization problem:

$$\min_w f(w) := \frac{1}{n}\sum_{i=1}^n \ell(w^\top u^{(i)}, v^{(i)}), \quad \text{subject to } \|w\|_0 \le k.$$

where $\ell(\cdot, \cdot)$ is a loss function measuring the discrepancy between prediction and response and $w$ is a set of parameters to be estimated. In the subsequent subsections, we will investigate sparse linear regression and sparse logistic regression as two popular examples of the above formulation.

### 3.1 Sparsity-constrained linear regression

Given a $\bar{k}$-sparse parameter vector $\bar{w}$, let us consider the samples are generated according to the linear model $v^{(i)} = \bar{w}^\top u^{(i)} + \varepsilon^{(i)}$ where $\varepsilon^{(i)}$ are $n$ i.i.d. sub-Gaussian random variables with

parameter $\sigma$. The sparsity-constrained least squared linear regression model is then given by:

$$\min_{w} f(w) = \frac{1}{2n} \sum_{i=1}^{n} \|v^{(i)} - w^{\top} u^{(i)}\|^2, \quad \text{subject to } \|w\|_0 \leq k. \tag{3}$$

Suppose $u^{(i)}$ are drawn from Gaussian distribution with covariance $\Sigma$. Then it holds with high probability that $f(w)$ has RSC constant $m_{2k} \geq \lambda_{\min}(\Sigma) - \mathcal{O}(k \log p/n)$ and RSS constant $M_{2k} \leq \lambda_{\max}(\Sigma) + \mathcal{O}(k \log p/n)$, and $\|\nabla f(\bar{w})\|_\infty = \mathcal{O}\left(\sigma\sqrt{\log p/n}\right)$. From Theorem 2 we know that for sufficiently large $n$, if the condition number $\lambda_{\max}(\Sigma)/\lambda_{\min}(\Sigma)$ is well bounded and $\bar{w}_{\min} > \mathcal{O}\left(\sigma\sqrt{k \log p/n}\right)$, then $\text{supp}(\bar{w})$ can be recovered by HTP after sufficient iteration. Since $\varepsilon^{(i)}$ are sub-Gaussian, we have $f(\bar{w}) = \frac{1}{2n} \sum_{i=1}^{n} \|\varepsilon^{(i)}\|^2 \leq \sigma^2$ holds with high probability. From Theorem 3 we can see that if $\bar{w}_{\min} > 1.62\sigma\sqrt{2/m_{2k}}$, then $\text{supp}(\bar{w})$ can be recovered, with high probability, by HTP with a sufficiently large sparsity level and a $\bar{k}$-sparse truncation postprocessing.

## 3.2 Sparsity-constrained logistic regression

Logistic regression is one of the most popular models in statistical learning. In this model the relation between the random feature vector $u \in \mathbb{R}^p$ and its associated random binary label $v \in \{-1, +1\}$ is determined by the conditional probability $\mathbb{P}(v|u; \bar{w}) = \exp(2v\bar{w}^{\top}u)/(1 + \exp(2v\bar{w}^{\top}u))$. Given a set of $n$ independently drawn data samples $\{(u^{(i)}, v^{(i)})\}_{i=1}^{n}$, the sparse logistic regression model learns the parameters $w$ so as to minimize the logistic log-likelihood over sparsity constraint:

$$\min_{w} f(w) = \frac{1}{n} \sum_{i=1}^{n} \log(1 + \exp(-2v^{(i)} w^{\top} u^{(i)})), \quad \text{subject to } \|w\|_0 \leq k. \tag{4}$$

It has been shown in (Bahmani et al., 2013, Corollary 1) that under mild conditions, $f(w)$ has RSC and RSS with overwhelming probability. Suppose $u^{(i)}$ are sub-Gaussian with parameter $\sigma$, then it is known from (Yuan et al., 2014) that $\|\nabla f(\bar{w})\|_\infty = \mathcal{O}\left(\sigma\sqrt{\log p/n}\right)$. Then from Theorem 2 we know that if the restrictive strong condition number is well bounded and $\bar{w}_{\min} > \mathcal{O}\left(\sigma\sqrt{k \log p/n}\right)$, then $\text{supp}(\bar{w})$ can be recovered by HTP after sufficient iteration. By using Theorem 3 and the fact $\sqrt{f(\bar{w}) - f(w^\star)} = \mathcal{O}(\sqrt{k}\|\nabla f(\bar{x})\|_\infty)$, we can show that if $\bar{w}_{\min} > \mathcal{O}\left(\sigma\sqrt{k \log p/n}\right)$, then with high probability, $\text{supp}(\bar{w})$ can be recovered by HTP using a sufficiently large sparsity level $k$ and proper postprocessing, without assuming bounded sparse condition number.

## 4 Numerical Results

In this section, we conduct a group of Monte-Carlo simulation experiments on sparse linear regression and sparse logistic regression models to verify our theoretical predictions.

**Data generation:** We consider a synthetic data model in which the sparse parameter $\bar{w}$ is a $p = 500$ dimensional vector that has $\bar{k} = 50$ nonzero entries drawn independently from the standard Gaussian distribution. Each data sample $u$ is a normally distributed dense vector. For the linear regression model, the responses are generated by $v = u\bar{w} + \varepsilon$ where $\varepsilon$ is a standard Gaussion noise. For the logistic regression model, the data labels, $v \in \{-1, 1\}$, are then generated randomly according to the Bernoulli distribution $\mathbb{P}(v = 1|u; \bar{w}) = \exp(2\bar{w}^{\top}u)/(1 + \exp(2\bar{w}^{\top}u))$. We allow the sample size $n$ to be varying and for each $n$, we generate 100 random copies of data independently.

**Evaluation metric:** In our experiment, we test HTP with varying sparsity level $k \geq \bar{k}$. We use two metrics to measure the support recovery performance. We say a *relaxed support recovery* is successful if $\text{supp}(\bar{w}) \subseteq \text{supp}(w^{(t)})$ and an *exact support recovery* is successful if $\text{supp}(\bar{w}) = \text{supp}(w^{(t)}, \bar{k})$. We replicate the experiment over the 100 trials and record the percentage of successful relaxed support recovery and percentage of successful exact support recovery for each configuration of $(n, k)$.

**Results:** Figure 1 shows the percentage of relaxed/exact success curves as functions of sample size $n$ under varying sparsity level $k$. From the curves in Figure 1(a) for the linear regression model we

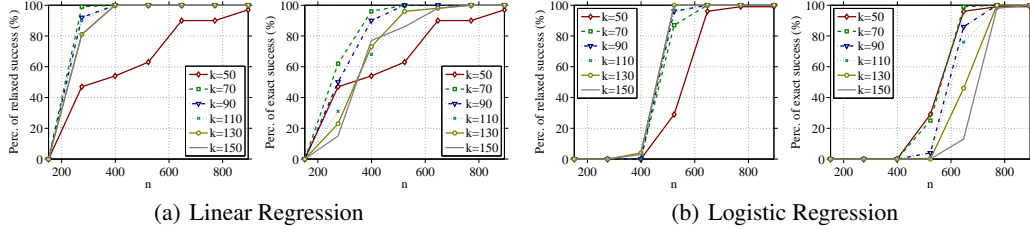

(a) Linear Regression            (b) Logistic Regression

Figure 1: Chance of relaxed success (left panel) and exact success (right panel) curves for linear regression and logistic regression models.

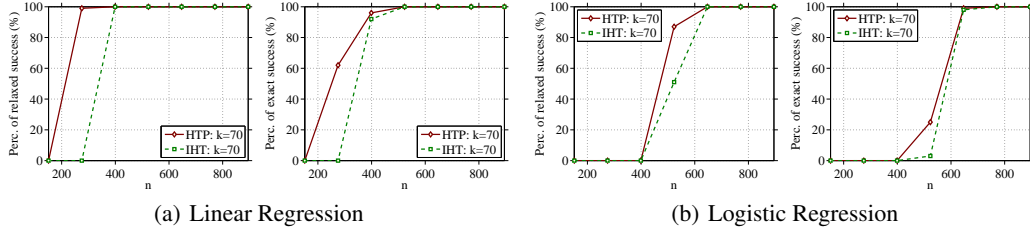

(a) Linear Regression            (b) Logistic Regression

Figure 2: HTP *versus* IHT: Chance of relaxed and exact success of support recovery.

can make two observations: 1) for each curve, the chance of success increases as $n$ increases, which matches the results in Theorem 1 and Theorem 2; 2) HTP has the best performance when using sparsity level $k = 70 > \bar{k}$. Also it can be seen that the percentage of relaxed success is less sensitive to $k$ than the percentage of exact success. These observations match the prediction in Theorem 3. Similar observations can be made from the curves in Figure 1(b) for the logistic regression model.

We have also compared HTP with IHT (Blumensath & Davies, 2009) in support recovery performance. Note that IHT is a simplified variant of HTP without the debiasing operation (S3) in Algorithm 1. Our exact support recovery analysis for HTP builds heavily upon such a debiasing operation. Figure 2 shows the chance of success curves for these two methods with sparsity level $k = 70$. Figure 2(a) shows that in linear regression model, HTP is superior to IHT when the sample size $n$ is relatively small and they become comparable as $n$ increases. Figure 2(b) indicates that HTP slightly outperforms IHT when applied to the considered logistic regression task. From this group of results we can draw the conclusion that the debiasing step of HPT does have significant impact on improving the support recovery performance especially in small sample size settings.

## 5   Conclusions

In this paper, we provided a deterministic support recovery analysis for HTP-style methods widely used in sparse learning. Theorem 1 establishes sufficient conditions for exactly recovering the global $k$-sparse minimizer $x^\star$ of the NP-hard problem (1). Theorem 2 provides sufficient conditions to guarantee the support recovery of an arbitrary $k$-sparse target solution. Theorem 3 further shows that even when the restricted strong condition number can be arbitrarily large, HTP is still able to recover a target sparse solution by using certain relaxed sparsity level in the algorithm. We have applied our deterministic analysis to sparse linear regression and sparse logistic regression to establish the sparsistency of HTP in these statistical learning models. Based on our theoretical justification and numerical observation, we conclude that HTP-style methods are not only accurate in parameter estimation, but also powerful for exactly recovering sparse signals even in noisy settings.

## Acknowledgments

Xiao-Tong Yuan and Ping Li were partially supported by NSF-Bigdata-1419210, NSF-III-1360971, ONR-N00014-13-1-0764, and AFOSR-FA9550-13-1-0137. Xiao-Tong Yuan is also partially supported by NSFC-61402232, NSFC-61522308, and NSFJP-BK20141003. Tong Zhang is supported by NSF-IIS-1407939 and NSF-IIS-1250985.

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
