[Supplementary Material]

# Abstract

This supplementary document contains the technical proofs of all the theorems in the NIPS'16 paper entitled "Exact Recovery of Hard Thresholding Pursuit". It is indeed the appendix section of the paper. The key technical lemmas are gathered in Appendix A, followed by the proofs of main results in Appendices B∼ E.

## A  Technical Lemmas

We present here a few technical lemmas to be used in our analysis.

**Lemma 1.** *Let $x$ be a $k$-sparse vector and $y = x - \eta \nabla f(x)$. If $f$ is $M_{2k}$-smooth, then the following inequality hods:*

$$f(y_k) \leq f(x) - \frac{1 - \eta M_{2k}}{2\eta} \|y_k - x\|^2.$$

*Proof.* Since $f$ is $M_{2k}$-smooth, it follows that

$$\begin{aligned} f(y_k) - f(x) &\leq \langle \nabla f(x), y_k - x \rangle + \frac{M_{2k}}{2} \|y_k - x\|^2 \\ &\overset{\xi_1}{\leq} -\frac{1}{2\eta} \|y_k - x\|^2 + \frac{M_{2k}}{2} \|y_k - x\|^2 \\ &= -\frac{1 - \eta M_{2k}}{2\eta} \|y_k - x\|^2, \end{aligned}$$

where $\xi_1$ follows from the fact that $y_k$ is the best $k$-support approximation to $y$ such that

$$\|y_k - y\|^2 = \|y_k - x + \eta \nabla f(x)\|^2 \leq \|x - x + \eta \nabla f(x)\|^2 = \|\eta \nabla f(x)\|^2,$$

which implies $2\eta \langle \nabla f(x), y_k - x \rangle \leq -\|y_k - x\|^2$. $\qquad\square$

**Lemma 2.** *Assume that $f$ is $m_s$-strongly convex. For any $\|x - x'\|_0 \leq s$ it holds that*

$$\|x - x'\| \leq \sqrt{\frac{2 \max \{f(x) - f(x'), 0\}}{m_s}} + \frac{2\|\nabla_{F \cup F'} f(x')\|}{m_s},$$

*where $F = supp(x)$ and $F' = supp(x')$.*

*Proof.* Since $f$ is $m_s$-strongly convex, we have

$$\begin{aligned} f(x) &\geq f(x') + \langle \nabla f(x'), x - x' \rangle + \frac{m_s}{2} \|x - x'\|^2 \\ &\geq f(x') - \|\nabla_{F \cup F'} f(x')\| \|x - x'\| + \frac{m_s}{2} \|x - x'\|^2, \end{aligned}$$

where the second inequality follows from Cauchy-Schwarz inequality. From this above inequality we can see that if $f(x) \leq f(x')$, then

$$\|x - x'\| \leq \frac{2\|\nabla_{F \cup F'} f(x')\|}{m_s}.$$

If otherwise $f(x) > f(x')$, then we have

$$\begin{aligned} \|x - x'\| &\leq \frac{\|\nabla_{F \cup F'} f(x')\| + \sqrt{\|\nabla_{F \cup F'} f(x')\|^2 + 2m_s(f(x) - f(x'))}}{m_s} \\ &\leq \frac{2\|\nabla_{F \cup F'} f(x')\| + \sqrt{2m_s(f(x) - f(x'))}}{m_s}. \end{aligned}$$

This proves the bound. $\qquad\square$

**Lemma 3.** *Assume that $f$ is $m_s$-strongly convex and $M_s$-smooth. For any index set $F$ with cardinality $|F| \leq s$ and any $x, y$ with $supp(x) \cup supp(y) \subseteq F$, if $\eta \in (0, 2m_s/M_s^2)$, then*

$$\|x - y - \eta \nabla_F f(x) + \eta \nabla_F f(y)\| \leq \sqrt{1 - 2\eta m_s + \eta^2 M_s^2} \|x - y\|,$$

*and $\sqrt{1 - 2\eta m_s + \eta^2 M_s^2} < 1$.*

*Proof.* By adding two copies of the inequality (2) with $x$ and $y$ interchanged and using the Theorem 2.1.5 in (Nesterov, 2004), we know that

$$(x - y)^\top (\nabla f(x) - \nabla f(y)) \geq m_s \|x - y\|^2, \quad \|\nabla_F f(x) - \nabla_F f(y)\| \leq M_s \|x - y\|.$$

For any $\eta > 0$ we have

$$\|x - y - \eta \nabla_F f(x) + \eta \nabla_F f(y)\|^2 \leq (1 - 2\eta m_s + \eta^2 M_s^2) \|x - y\|^2.$$

Obviously $1 - 2\eta m_s + \eta^2 M_s^2 \geq 1 - m_s^2/M_s^2 \geq 0$, and $\eta < 2m_s/M_s^2$ implies $\sqrt{1 - 2\eta m_s + \eta^2 M_s^2} < 1$. This proves the lemma. $\square$

**Lemma 4.** *Assume that $f$ is $M_s$-smooth and $m_s$-strongly convex. Let $F$ and $F'$ be two index sets with cardinality $|F \cup F'| = s$. Let $x = \arg\min_{supp(y) \subseteq F} f(y)$ and $supp(x') \subseteq F'$. Then for any $\eta \in (0, 2m_s/M_s^2)$, the following two inequalities hold*

$$\|(x - x')_F\| \leq \frac{\rho \|x'_{F' \setminus F}\|}{1 - \rho} + \frac{\eta \|\nabla_{F \cup F'} f(x')\|}{(1 - \rho)}, \tag{A.1}$$

$$\|x - x'\| \leq \frac{\|x'_{F' \setminus F}\|}{1 - \rho} + \frac{\eta \|\nabla_{F \cup F'} f(x')\|}{(1 - \rho)}, \tag{A.2}$$

*where $\rho = \sqrt{1 - 2\eta m_s + \eta^2 M_s^2} < 1$.*

*Proof.* Since $x$ is the minimum of $f(y)$ restricted over the supporting set $F$, we have $\langle \nabla f(x), z \rangle = 0$ whenever $supp(z) \subseteq F$. It follows that

$$
\begin{aligned}
&\|(x - x')_F\|^2 \\
=& \langle x - x', (x - x')_F \rangle \\
=& \langle x - x' - \eta \nabla_{F \cup F'} f(x) + \eta \nabla_{F \cup F'} f(x'), (x - x')_F \rangle - \eta \langle \nabla_{F \cup F'} f(x'), (x - x')_F \rangle \\
\overset{\xi_1}{\leq}& \sqrt{1 - 2\eta m_s + \eta^2 M_s^2} \|x - x'\| \|(x - x')_F\| + \eta \|\nabla_{F \cup F'} f(x')\| \|(x - x')_F\|,
\end{aligned}
$$

where $\xi_1$ uses Lemma 3. Let us abbreviate $\rho = \sqrt{1 - 2\eta m_s + \eta^2 M_s^2}$. After simplification, we have

$$\|(x - x')_F\| \leq \rho \|x - x'\| + \eta \|\nabla_{F \cup F'} f(x')\|. \tag{A.3}$$

It follows that

$$
\begin{aligned}
\|x - x'\| \leq& \|(x - x')_F\| + \|(x - x')_{F' \setminus F}\| \\
\leq& \rho \|x - x'\| + \eta \|\nabla_{F \cup F'} f(x')\| + \|(x - x')_{F' \setminus F}\|.
\end{aligned}
$$

After rearrangement we obtain

$$
\begin{aligned}
\|x - x'\| \leq& \frac{\|(x - x')_{F' \setminus F}\|}{1 - \rho} + \frac{\eta \|\nabla_{F \cup F'} f(x')\|}{1 - \rho} \\
=& \frac{\|x'_{F' \setminus F}\|}{1 - \rho} + \frac{\eta \|\nabla_{F \cup F'} f(x')\|}{1 - \rho}.
\end{aligned} \tag{A.4}
$$

By combining (A.3) and (A.4) we get

$$\|(x - x')_F\| \leq \frac{\rho \|x'_{F' \setminus F}\|}{1 - \rho} + \frac{\eta \|\nabla_{F \cup F'} f(x')\|}{1 - \rho}.$$

This proves the desired bounds in this lemma. $\square$

# B Proof of Proposition 1

The following lemma gives a necessary condition on the value $x_{\min}^\star$.

**Lemma 5.** *If $f$ is $M_{2k}$-smooth, then the following inequality holds for the global minimizer $x^\star$:*

$$x_{\min}^\star \geq \frac{\|\nabla f(x^\star)\|_\infty}{M_{2k}}.$$

*Proof.* Assume otherwise that $\vartheta^\star := \frac{M_{2k} x_{\min}^\star}{\|\nabla f(x^\star)\|_\infty} < 1$. Let us consider $\tilde{x}^\star = x^\star - \eta \nabla f(x^\star)$ with any $\eta \in (\vartheta^\star/M_{2k}, 1/M_{2k})$. From Lemma 1 we get that

$$f(\tilde{x}_k^\star)) \leq f(x^\star) - \frac{1 - \eta M_{2k}}{2\eta} \|\tilde{x}_k^\star - x^\star\|^2.$$

Since $\eta < \frac{1}{M_{2k}}$ and $x_{\min}^\star = \frac{\vartheta^\star \|\nabla f(x^\star)\|_\infty}{M_{2k}} < \eta \|\nabla f(x^\star)\|_\infty$, we have $\tilde{x}_k^\star \neq x^\star$ and thus it follows from the above inequality that $f(\tilde{x}_k^\star) < f(x^\star)$ which contradicts the optimality of $x^\star$. $\square$

We also need the following lemma in our proof.

**Lemma 6.** *Assume that $f$ is $M_{2k}$-smooth and $m_{2k}$-strongly convex. Let $\bar{x}$ be an arbitrary $k$-sparse vector satisfying $\bar{x} = \arg\min_{supp(x) \subseteq supp(\bar{x})} f(x)$ and $\bar{x}_{\min} \geq \frac{\bar{\vartheta}}{M_{2k}} \|\nabla f(\bar{x})\|_\infty$ for some $\bar{\vartheta} > 1$. If $\frac{m_{2k}}{M_{2k}} \geq \max\left\{\frac{3\bar{\vartheta}+1}{4\bar{\vartheta}}, \frac{\sqrt{3}}{2}\right\}$, then $\bar{x}$ is the global minimizer, i.e., $\bar{x} = x^\star$.*

*Proof.* Denote $\bar{F} = supp(\bar{x})$. Assume otherwise that $\bar{F} \neq F^\star$. By using the inequality (A.1) in Lemma 4 and the definition of $\bar{\vartheta}$ we have

$$
\begin{aligned}
\|x_{F^\star \setminus \bar{F}}^\star\| &\leq \|(x^\star - \bar{x})_{F^\star}\| \\
&\leq \frac{\rho \|\bar{x}_{\bar{F} \setminus F^\star}\|}{1 - \rho} + \frac{m_{2k} \|\nabla_{F^\star \setminus \bar{F}} f(\bar{x})\|}{(1 - \rho) M_{2k}^2} \\
&\leq \frac{(\bar{\vartheta}\rho + m_{2k}/M_{2k}) \|\bar{x}_{\bar{F} \setminus F^\star}\|}{\bar{\vartheta}(1 - \rho)} \\
&\overset{\xi_1}{\leq} \left(1 + \frac{2m_{2k}}{\bar{\vartheta} M_{2k}}\right) \|\bar{x}_{\bar{F} \setminus F^\star}\|,
\end{aligned}
\tag{A.5}
$$

where the last inequality $\xi_1$ follows from $\rho \leq 0.5$ (as $m_{2k}/M_{2k} \geq \frac{\sqrt{3}}{2}$). Let $F = F^\star \cup \bar{F}$. Since $f$ is $m_{2k}$-strongly convex, it can be easily verified that

$$\|\nabla_F f(\bar{x}) - \nabla_F f(x^\star)\| \geq m_{2k} \|\bar{x} - x^\star\|. \tag{A.6}$$

It follows that

$$
\begin{aligned}
&\|\nabla_{F^\star \setminus \bar{F}} f(\bar{x})\| + \|\nabla_{\bar{F} \setminus F^\star} f(x^\star)\| \\
&\overset{\xi_1}{=} \|\nabla_F f(\bar{x})\| + \|\nabla_F f(x^\star)\| \\
&\overset{\xi_2}{\geq} \|\nabla_F f(\bar{x}) - \nabla_F f(x^\star)\| \\
&\overset{\xi_3}{\geq} m_{2k} \|\bar{x} - x^\star\| \\
&\geq m_{2k} \left(\|\bar{x}_{\bar{F} \setminus F^\star}\| + \|x_{F^\star \setminus \bar{F}}^\star\|\right),
\end{aligned}
\tag{A.7}
$$

where the inequality $\xi_1$ follows from the optimality condition, i.e., $\nabla_{F^\star} f(x^\star) = \nabla_{\bar{F}} f(\bar{x}) = 0$; the inequality $\xi_2$ follows from triangle inequality; the inequality $\xi_3$ follows from (A.6). Since the

definition of $\bar{\vartheta}$ implies $\|\nabla_{F^\star\setminus\bar{F}}f(\bar{x})\| \le \frac{M_{2k}}{\bar{\vartheta}}\|\bar{x}_{\bar{F}\setminus F^\star}\|$, it follows from the inequality (A.7) that

$$\|\nabla_{\bar{F}\setminus F^\star}f(x^\star)\| \ge \left(m_{2k} - \frac{M_{2k}}{\bar{\vartheta}}\right)\|\bar{x}_{\bar{F}\setminus F^\star}\| + m_{2k}\|x^\star_{F^\star\setminus\bar{F}}\|$$

$$\overset{\xi_1}{\ge} \left(\frac{m_{2k} - M_{2k}/\bar{\vartheta}}{1 + 2m_{2k}/(\bar{\vartheta}M_{2k})} + m_{2k}\right)\|x^\star_{F^\star\setminus\bar{F}}\|$$

$$\overset{\xi_2}{>} \left(\frac{m_{2k} - M_{2k}/\bar{\vartheta}}{3} + m_{2k}\right)\|x^\star_{F^\star\setminus\bar{F}}\|$$

$$= \left(\frac{4m_{2k} - M_{2k}/\bar{\vartheta}}{3}\right)\|x^\star_{F^\star\setminus\bar{F}}\|$$

$$\overset{\xi_3}{\ge} M_{2k}\|x^\star_{F^\star\setminus\bar{F}}\|,$$

where inequality $\xi_1$ follows from (A.5) and the assumption on $m_{2k}/M_{2k}$ which implies $m_{2k} > \frac{M_{2k}}{\bar{\vartheta}}$, $\xi_2$ follows from $\bar{\vartheta} > 1$ and $m_{2k} < M_{2k}$, and $\xi_3$ follows from $m_{2k}/M_{2k} \ge \frac{3\bar{\vartheta}+1}{4\bar{\vartheta}}$. The previous inequality obviously leads to

$$x^\star_{\min} < \frac{\|\nabla f(x^\star)\|_\infty}{M_{2k}}.$$

This contradicts the x-min condition we have proved for $x^\star$ in Lemma 5. Therefore, we must have $\bar{x} = x^\star$ holds. $\qquad\square$

Now we can prove Proposition 1.

*Proof of Proposition 1.* We first show that $\text{supp}(\bar{x}) = \text{supp}(x^\star)$ if Condition (1) holds. Assume otherwise $\text{supp}(\bar{x}) \ne \text{supp}(x^\star)$. From the optimality of $x^\star$ we have $f(x^\star) \le f(\bar{x})$. By invoking Lemma 2 we get

$$\bar{x}_{\min} < \|x^\star - \bar{x}\| \le \frac{2\sqrt{2k}\|\nabla f(\bar{x})\|_\infty}{m_{2k}},$$

which contradicts the condition.

Next we show that $\text{supp}(\bar{x}) = \text{supp}(x^\star)$ if Condition (2) is satisfied. From the definition of $\bar{x}^\star$ and $\bar{l}$ we have

$$f(\bar{x}) \ge f(\bar{x}^\star)$$

$$\ge f(\bar{x}) - \|\nabla f(\bar{x})\|_\infty\|\bar{x}^\star - \bar{x}\|_1 + \frac{\bar{l}}{2}\|\bar{x}^\star - \bar{x}\|_1^2$$

$$\ge f(\bar{x}) - \frac{\|\nabla f(\bar{x})\|_\infty^2}{\bar{l}} - \frac{\bar{l}}{4}\|\bar{x}^\star - \bar{x}\|_1^2 + \frac{\bar{l}}{2}\|\bar{x}^\star - \bar{x}\|_1^2$$

$$= f(\bar{x}) - \frac{\|\nabla f(\bar{x})\|_\infty^2}{\bar{l}} + \frac{\bar{l}}{4}\|\bar{x}^\star - \bar{x}\|_1^2.$$

Therefore,

$$\|\bar{x}^\star - \bar{x}\|_1 \le \frac{2\|\nabla f(\bar{x})\|_\infty}{\bar{l}}. \tag{A.8}$$

Let $\bar{F} = \text{supp}(\bar{x})$. Since $f$ is $M_{2k}$-smooth, we know from (Nesterov, 2004, Theorem 2.1.5) and the above inequality that

$$\|\nabla f(\bar{x}^\star) - \nabla f(\bar{x})\| \le M_{2k}\|\bar{x}^\star - \bar{x}\| \le M_{2k}\|\bar{x}^\star - \bar{x}\|_1 \le \frac{2M_{2k}\|\nabla f(\bar{x})\|_\infty}{\bar{l}}, \tag{A.9}$$

where we have also used the fact $\|x\| \le \|x\|_1$. By applying triangle inequality to (A.8) and (A.9) we further get

$$\bar{x}^\star_{\min} \ge \bar{x}_{\min} - \frac{2\|\nabla f(\bar{x})\|_\infty}{\bar{l}}, \quad \|\nabla f(\bar{x}^\star)\|_\infty \le \frac{(\bar{l} + 2M_{2k})\|\nabla f(\bar{x})\|_\infty}{\bar{l}}.$$

Given the condition on $\bar{x}_{\min}$, we can obtain from the above

$$\bar{x}^\star_{\min} \ge \frac{\bar{\vartheta}\|\nabla f(\bar{x}^\star)\|_\infty}{M_{2k}}.$$

Thus, by using Lemma 6 we obtain that $\bar{x}^\star = x^\star$. This leads the desired result since $\text{supp}(\bar{x}^\star) = \text{supp}(\bar{x})$. $\qquad\square$

## C  Proof of Theorem 1

In large picture, the proof of Theorem 1 contains the following three key steps:

(a) In the first step, we prove that under the conditions stated in the theorem, HTP terminates when $x^{(t)}$ reaches $x^\star$.

(b) In the second step, we show that if $x^{(t-1)} \neq x^\star$, then HTP will output $x^{(t)} \neq x^{(t-1)}$. That is, the algorithm will not terminate before reaching $x^\star$.

(c) In the final step, we show that the conditions in the theorem guarantee a unique optimal solution $x^\star$ and finite termination of HTP.

*Proof of Theorem 1.* **Step (a)**: We first prove that under the conditions stated in the theorem, the algorithm will terminate when it reaches $x^\star$. Indeed, let us assume $x^{(t-1)} = x^\star$. Since $\vartheta^\star > 1$ and $\eta = m_{2k}/M_{2k}^2 < 1/M_{2k}$, we obtain that

$$x_{\min}^{(t-1)} = x_{\min}^\star = \frac{\theta^\star \|\nabla f(x^\star)\|_\infty}{M_{2k}} > \frac{\|\nabla f(x^\star)\|_\infty}{M_{2k}} > \eta \|\nabla f(x^\star)\|_\infty = \eta \nabla_{\max}^{(t-1)},$$

which indicates that the top $k$ (in magnitude) entries of $\tilde{x}^{(t)} = x^{(t-1)} - \eta \nabla f(x^{(t-1)})$ are exactly the $k$ nonzero entries of $x^{(t-1)}$ and thus $F^{(t)} = F^{(t-1)}$. The step **S3** of HTP ensures $x^{(t)} = x^{(t-1)}$. Therefore, HTP terminates at $x^{(t-1)} = x^\star$.

**Step (b)**: Further, we show that at time instance $t - 1$, if $x^{(t-1)} \neq x^\star$, then HTP will output $x^{(t)} \neq x^{(t-1)}$. From the inequality (A.1) and the definition of $\vartheta^\star$ we have

$$
\begin{aligned}
\|x_{F^{(t-1)}\setminus F^\star}^{(t-1)}\| &\leq \|(x^{(t-1)} - x^\star)_{F^{(t-1)}}\| \\
&\leq \frac{\rho \|(x_{F^\star \setminus F^{(t-1)}}^\star\|}{1 - \rho} + \frac{\eta \|\nabla_{F^{(t-1)}\setminus F^\star} f(x^\star)\|}{1 - \rho} \\
&\leq \frac{(\rho \vartheta^\star + \eta M_{2k}) \|x_{F^\star \setminus F^{(t-1)}}^\star\|}{\vartheta^\star (1 - \rho)} \\
&\stackrel{\xi_1}{\leq} \left(1 + \frac{2m_{2k}}{\vartheta^\star M_{2k}}\right) \|x_{F^\star \setminus F^{(t-1)}}^\star\|,
\end{aligned}
\tag{A.10}
$$

where the last inequality $\xi_1$ follows from $\eta = m_{2k}/M_{2k}^2$ and $\rho \leq 0.5$ (as $m_{2k}/M_{2k} \geq \frac{7\vartheta^\star + 1}{8\vartheta^\star} > \frac{\sqrt{3}}{2}$). Let $F = F^\star \cup F^{(t-1)}$. Since $f$ is $m_{2k}$-strongly convex, it can be easily verified that

$$\|\nabla_F f(x^\star) - \nabla_F f(x^{(t-1)})\| \geq m_{2k} \|x^\star - x^{(t-1)}\|. \tag{A.11}$$

Thus

$$
\begin{aligned}
&\|\nabla_{F^{(t-1)}\setminus F^\star} f(x^\star)\| + \|\nabla_{F^\star \setminus F^{(t-1)}} f(x^{(t-1)})\| \\
&\stackrel{\xi_1}{=} \|\nabla_F f(x^\star)\| + \|\nabla_F f(x^{(t-1)})\| \\
&\stackrel{\xi_2}{\geq} \|\nabla_F f(x^\star) - \nabla_F f(x^{(t-1)})\| \\
&\stackrel{\xi_3}{\geq} m_{2k} \|x^\star - x^{(t-1)}\| \\
&\geq m_{2k} \left(\|x_{F^\star \setminus F^{(t-1)}}^\star\| + \|x_{F^{(t-1)}\setminus F^\star}^{(t-1)}\|\right),
\end{aligned}
\tag{A.12}
$$

where the inequality $\xi_1$ follows from the optimality condition of $\nabla_{F^\star} f(x^\star) = \nabla_{F^{(t-1)}} f(x^{(t-1)}) = 0$; the inequality $\xi_2$ follows from triangle inequality; the inequality $\xi_3$ follows from (A.11). Since the definition of $\vartheta^\star$ implies $\|\nabla_{F^{(t-1)}\setminus F^\star} f(x^\star)\| \leq \frac{M_{2k}}{\vartheta^\star} \|x_{F^\star \setminus F^{(t-1)}}^\star\|$, it follows from the inequal-

ity (A.12) that

$$\|\nabla_{F^\star \setminus F^{(t-1)}} f(x^{(t-1)})\| \geq \left( m_{2k} - \frac{M_{2k}}{\vartheta^\star} \right) \|x^\star_{F^\star \setminus F^{(t-1)}}\| + m_{2k} \|x^{(t-1)}_{F^{(t-1)} \setminus F^\star}\|$$

$$\overset{\xi_1}{\geq} \left( \frac{m_{2k} - M_{2k}/\vartheta^\star}{1 + 2m_{2k}/(\vartheta^\star M_{2k})} + m_{2k} \right) \|x^{(t-1)}_{F^{(t-1)} \setminus F^\star}\|$$

$$\overset{\xi_2}{>} \left( \frac{m_{2k} - M_{2k}/\vartheta^\star}{3} + m_{2k} \right) \|x^{(t-1)}_{F^{(t-1)} \setminus F^\star}\|$$

$$= \left( \frac{4m_{2k} - M_{2k}/\vartheta^\star}{3} \right) \|x^{(t-1)}_{F^{(t-1)} \setminus F^\star}\|,$$

where inequality $\xi_1$ follows from (A.10) and the assumption on $m_{2k}/M_{2k}$ which implies $m_{2k} > \frac{M_{2k}}{\vartheta^\star}$, and $\xi_2$ follows from $\vartheta^\star > 1$ and $m_{2k} < M_{2k}$. Now we claim that $\eta \|\nabla_{F^\star \setminus F^{(t-1)}} f(x^{(t-1)})\| > \|x^{(t-1)}_{F^{(t-1)} \setminus F^\star}\|$. Indeed, since $\eta = \frac{m_{2k}}{M_{2k}^2}$ and $\frac{m_{2k}}{M_{2k}} \geq \frac{1+7\vartheta^\star}{8\vartheta^\star}$, it can be verified from the previous inequality that

$$\eta \|\nabla_{F^\star \setminus F^{(t-1)}} f(x^{(t-1)})\| > \left( \frac{4m_{2k}^2}{3M_{2k}^2} - \frac{m_{2k}}{3M_{2k}\vartheta^\star} \right) \|x^{(t-1)}_{F^{(t-1)} \setminus F^\star}\| \geq \|x^{(t-1)}_{F^{(t-1)} \setminus F^\star}\|.$$

Thus at least the nonzero entries of $x^{(t-1)}$ supported on $F^{(t-1)} \setminus F^\star$ and the entries of $\nabla f(x^{(t-1)})$ supported on $F^\star \setminus F^{(t-1)}$ can (but not have to) be swapped in the step **S2**. Thus $F^{(t)} \neq F^{(t-1)}$ and HTP proceeds to $x^{(t)} \neq x^{(t-1)}$.

**Step (c)**: The final step is to further show that under the conditions stated in the theorem, the optimal solution $x^\star$ is unique and the sequence $\{f(x^{(t)})\}$ generated by HTP reach $f(x^\star)$ within a finite number of iteration. We first prove the uniqueness of optimal solution by contradiction. Assume otherwise there exists $x^{\star\star} \neq x^\star$ such that $f(x^{\star\star}) = f(x^\star)$. Let $F^{\star\star} = \text{supp}(x^{\star\star})$. Since $f$ is $m_{2k}$-strongly convex, it is true that $F^{\star\star} \neq F^\star$. Similar to the arguments in part(b) we can prove that

$$\eta \|\nabla_{F^\star \setminus F^{\star\star}} f(x^{\star\star})\| > \|x^{\star\star}_{F^{\star\star} \setminus F^\star}\|.$$

Now consider $\tilde{x}^{\star\star} = x^{\star\star} - \eta \nabla f(x^{\star\star})$. From the above inequality we get $\tilde{x}^{\star\star}_k \neq x^{\star\star}$. By using Lemma 1 we get

$$f(\tilde{x}^{\star\star}_k) - f(x^\star) = f(\tilde{x}^{\star\star}_k) - f(x^{\star\star}) \leq -\frac{1 - \eta M_{2k}}{2\eta} \|\tilde{x}^{\star\star}_k - x^{\star\star}\|^2 < 0.$$

This contradicts the optimality of $x^\star$. Therefore, the optimal solution $x^\star$ is unique under the conditions in the theorem.

Last, we show that $f(x^{(t)})$ will reach $f(x^\star)$ in finite steps. In the above Step (b) we have proved $x^{(t)} \neq x^{(t-1)}$ whenever $x^{(t)} \neq x^\star$, and in such a case it must hold that $|F^\star \setminus F^{(t-1)}| \leq |F^{(t)} \setminus F^{(t-1)}|$. Thus from the fact that $\nabla_{F^{(t)} \setminus F^{(t-1)}} f(x^{(t-1)})$ contains the top $|F^{(t)} \setminus F^{(t-1)}|$ entries of $\nabla f(x^{(t-1)})$ we get

$$\|\nabla_{F^\star \setminus F^{(t-1)}} f(x^{(t-1)})\| \leq \|\nabla_{F^{(t)} \setminus F^{(t-1)}} f(x^{(t-1)})\|.$$

From the definition of $\tilde{x}^{(t)}$ we have that the following inequality holds:

$$\|\tilde{x}^{(t)}_{F^{(t)}} - x^{(t-1)}\| \geq \eta \|\nabla_{F^{(t)} \setminus F^{(t-1)}} f(x^{(t-1)})\|. \tag{A.13}$$

By using Lemma 1, (A.13) and the above inequality we have

$$f(x^{(t)}) - f(x^{(t-1)}) \leq f(\tilde{x}^{(t)}_{F^{(t)}}) - f(x^{(t-1)}) \leq -\frac{(1 - \eta M_{2k})\eta}{2} \|\nabla_{F^\star \setminus F^{(t-1)}} f(x^{(t-1)})\|^2. \tag{A.14}$$

From the $m_{2k}$-strong convexity of $f$ we have

$$\frac{m_{2k}}{2} \|x^\star - x^{(t-1)}\|^2 \leq f(x^\star) - f(x^{(t-1)}) - (x^\star - x^{(t-1)})^\top \nabla f(x^{(t-1)})$$

$$\overset{\xi_1}{\leq} f(x^\star) - f(x^{(t-1)}) + \frac{m_{2k}}{2} \|x^\star - x^{(t-1)}\|^2 + \frac{1}{2m_{2k}} \|\nabla_{F^\star \setminus F^{(t-1)}} f(x^{(t-1)})\|^2,$$

where $\xi_1$ follows from Cauchy-Schwartz inequality, $ma^2/2 + b^2/(2m) \geq ab$ for any $m > 0$, and $\nabla_{F^{(t-1)}} f(x^{(t-1)}) = 0$. This implies

$$\|\nabla_{F^\star \backslash F^{(t-1)}} f(x^{(t-1)})\|^2 \geq 2m_{2k}\left[f(x^{(t-1)}) - f(x^\star)\right]. \tag{A.15}$$

By combining (A.14) and (A.15) we arrive at

$$f(x^{(t)}) - f(x^{(t-1)}) \leq -(1-\eta M_{2k})m_{2k}\eta(f(x^{(t-1)}) - f(x^\star)) = -\left(1 - \frac{m_{2k}}{M_{2k}}\right)\frac{m_{2k}^2}{M_{2k}^2}(f(x^{(t-1)}) - f(x^\star)).$$

Therefore, we get

$$f(x^{(t)}) - f(x^\star) \leq (1-\nu)(f(x^{(t-1)}) - f(x^\star)),$$

where

$$\nu = \left(1 - \frac{m_{2k}}{M_{2k}}\right)\frac{m_{2k}^2}{M_{2k}^2}.$$

Therefore $f(x^{(t)}) - f(x^\star) \leq \triangle^{-\star}$ when $t \geq \frac{1}{\nu}\ln\frac{\triangle^{(0)}}{\triangle^{-\star}}$ (note that $\triangle^{-\star} > 0$ due to the uniqueness of optimal solution). After that, we have $f(x^{(t)}) < f(x^{-\star})$ and thus $f(x^{(t)}) = f(x^\star)$. Since $x^\star$ is unique, it must hold that $x^{(t)} = x^\star$.

By combining the above three steps (a), (b) and (c), we complete the proof of Theorem 1. $\qquad\square$

## D   Proof of Theorem 2

*Proof.* Since the conditions of Theorem 1 are assumed to be fulfilled, it follows from the theorem that $x^{(t)} = x^\star$ after sufficient iteration. If $\bar{x}_{\min} \geq \frac{2\sqrt{2k}}{m_{2k}}\|\nabla f(\bar{x})\|_\infty$, then we know from Proposition 1 that $\text{supp}(\bar{x}) = \text{supp}(x^\star)$, and thus $\text{supp}(x^{(t)}) = \text{supp}(\bar{x})$. If $\bar{x}_{\min} \geq \left(\frac{\vartheta^\star}{M_{2k}} + \frac{2\vartheta^\star+2}{l}\right)\|\nabla f(\bar{x})\|_\infty$, then using the arguments in the proof of Proposition 1 we can derive that

$$\bar{x}_{\min}^\star \geq \frac{\vartheta^\star\|\nabla f(\bar{x}^\star)\|_\infty}{M_{2k}}.$$

Since $\frac{m_{2k}}{M_{2k}} \geq \frac{7\vartheta^\star+1}{8\vartheta^\star}$, we get from Lemma 6 that $\bar{x}^\star = x^\star$. This leads to the desired result as $\text{supp}(x^{(t)}) = \text{supp}(x^\star) = \text{supp}(\bar{x}^\star) = \text{supp}(\bar{x})$. $\qquad\square$

## E   Proof of Theorem 3

*Proof of Theorem 3.* **Part(a)**. This part can be proved by showing that $F^{(t)} \neq F^{(t-1)}$ whenever $\text{supp}(\bar{x}) \not\subseteq \text{supp}(x^{(t-1)})$. To this end, let us assume $\text{supp}(\bar{x}) \not\subseteq \text{supp}(x^{(t-1)})$. Then

$$\bar{x}_{\min} + \|x_{F^{(t-1)}\backslash\bar{F}}^{(t-1)}\| \leq \|\bar{x} - x^{(t-1)}\|$$

$$\overset{\xi_1}{\leq} \sqrt{\frac{2\max\left\{f(\bar{x}) - f(x^{(t-1)}), 0\right\}}{m_{2k}}} + \frac{2\|\nabla_{\bar{F}\backslash F^{(t-1)}} f(x^{(t-1)})\|}{m_{2k}}$$

$$\overset{\xi_2}{\leq} \sqrt{\frac{2(f(\bar{x}) - f(x^\star))}{m_{2k}}} + \frac{2\|\nabla_{\bar{F}\backslash F^{(t-1)}} f(x^{(t-1)})\|}{m_{2k}},$$

where "$\xi_1$" follows from Lemma 2 and "$\xi_2$" is due to the fact of $f(x^{(t-1)}) \geq f(x^\star)$. Since it is assumed $\bar{x}_{\min} > \sqrt{\frac{2(f(\bar{x})-f(x^\star))}{m_{2k}}}$, the above inequality implies that

$$\|x_{F^{(t-1)}\backslash\bar{F}}^{(t-1)}\| < \frac{2\|\nabla_{\bar{F}\backslash F^{(t-1)}} f(x^{(t-1)})\|}{m_{2k}}.$$

Let $x_{bot(|\bar{F}\backslash F^{(t-1)}|)}^{(t-1)}$ be the smallest (in magnitude) $|\bar{F}\backslash F^{(t-1)}|$ elements of $x^{(t-1)}$. Then it can be shown from the previous inequality that

$$\|\nabla_{\bar{F}\backslash F^{(t-1)}} f(x^{(t-1)})\| > \frac{m_{2k}|F^{(t-1)}\backslash\bar{F}|^{1/2}}{2|\bar{F}\backslash F^{(t-1)}|^{1/2}}\|x_{bot(|\bar{F}\backslash F^{(t-1)}|)}^{(t-1)}\| \geq \frac{(k-\bar{k})^{1/2}m_{2k}}{2\bar{k}^{1/2}}\|x_{bot(|\bar{F}\backslash F^{(t-1)}|)}^{(t-1)}\|.$$

Since $\eta = \frac{1}{2M_{2k}}$ and $k \geq \left(1 + \frac{16M_{2k}^2}{m_{2k}^2}\right)\bar{k}$, we thus have

$$\eta\|\nabla_{\bar{F}\backslash F^{(t-1)}}f(x^{(t-1)})\| > \|x_{bot(|\bar{F}\backslash F^{(t-1)}|)}^{(t-1)}\|.$$

This further implies at least the bottom $|\bar{F}\backslash F^{(t-1)}|$ entries of $x^{(t-1)}$ can (but not have to) be swapped in the step **S2** of Algorithm 1, and thus $F^{(t)} \neq F^{(t-1)}$. Therefore, when the algorithm terminates at time instance $t$, i.e., $F^{(t+1)} = F^{(t)}$, we must have $\mathrm{supp}(\bar{x}) \subseteq \mathrm{supp}(x^{(t)})$ holds.

Finally, we claim that HTP is finite under the assumed conditions. Indeed, based on Lemma 1 it is easy to verify that when $\eta = \frac{1}{M_{2k}}$, the sequence $\{f(x^{(t)})\}$ generated by Algorithm 1 is monotone. Since the number of $k$-support index sets is finite, the sequence $\{f(x^{(t)})\}$ will be eventually periodic, and thus must be eventually a constant. Therefore we deduce that $\tilde{x}_k^{(t)} = x^{(t-1)}$, i.e., $F^{(t)} = F^{(t-1)}$, for $t$ large enough. This completes the proof of Part (a).

**Part(b)**: Given the conditions in this part, we know from Part(a) that $\mathrm{supp}(\bar{x}) \subseteq \mathrm{supp}(x^{(t)})$ when HTP terminates at time instance $t$. Let us assume $\mathrm{supp}(\bar{x}) \neq \mathrm{supp}(x_{\bar{k}}^{(t)})$. Then

$$\begin{aligned}
\bar{x}_{\min} &\leq \|\bar{x} - x_{\bar{k}}^{(t)}\| \\
&\overset{\xi_1}{\leq} 1.62\|\bar{x} - x^{(t)}\| \\
&\overset{\xi_2}{\leq} 1.62\sqrt{\frac{2(f(\bar{x}) - f(x^\star))}{m_{2k}}} + \frac{2\|\nabla_{\bar{F}\backslash F^{(t)}}f(x^{(t)})\|}{m_{2k}} \\
&\overset{\xi_3}{=} 1.62\sqrt{\frac{2(f(\bar{x}) - f(x^\star))}{m_{2k}}},
\end{aligned}$$

where "$\xi_1$" is based on the truncation error bound in (Shen & Li, 2016, Theorem 1), "$\xi_2$" follows from Lemma 2 and the fact of $f(x^{(t)}) \geq f(x^\star)$, and "$\xi_3$" is the consequence of $\bar{F} \subseteq F^{(t)}$ and the optimality of $x^{(t)}$ over $F^{(t)}$. This above inequality contradicts the assumption on $\bar{x}_{\min}$. Therefore, it must holds that $\mathrm{supp}(\bar{x}) = \mathrm{supp}(x_{\bar{k}}^{(t)})$. $\qquad\square$