[Reviews · NeurIPS 2016]

Reviewer 1

Summary

This paper derives guarantees for a Hard Thresholding Pursuit, a greedy method for sparse-estimation problems, which hold under restricted strong convexity and smoothness assumptions.

Qualitative Assessment

This paper provides a precise theoretical analysis of the support-detection performance of Hard Thresholding Pursuit (HTP), an iterative method for sparse estimation in high-dimensional statistics and signal processing. The paper is well written and explains the authors' contributions with respect to other work on HTP very carefully. The numerical simulations, however, are rather limited and hardly warrant the statement that "Numerical results on simulated data confirms (sic) our theoretical predictions.". A weakness of the paper is that it does not compare HTP to other algorithms (except for a very brief mention of an l1-norm regularized estimator at the end of Section 2) either theoretically or numerically. In particular, the authors apply their general results to linear regression and sparse logistic regression, but do not provide a context for their conclusions. Despite this, I think that the results are interesting.

Confidence in this Review

2-Confident (read it all; understood it all reasonably well)


Reviewer 2

Summary

The paper proposes a new analysis of sparse recovery conditions for Hard Thresholding Pursuit, describes its implications for two regression problems, and provides a short numerical illustration of the empirical performance of HTP on synthetic regression problems.

Qualitative Assessment

See fatal flaw.

Confidence in this Review

3-Expert (read the paper in detail, know the area, quite certain of my opinion)


Reviewer 3

Summary

The paper considers the class of hard threshold pursuit for the general high dimensional problems. To my best knowledge, this is the first attempt to provide the general theorem on the support set recovery for the HTP-type algorithms. It provides three variants of theorems to cover different situations: i) the support recovery for the global minimizer, ii) the support recovery for an arbitrary sparse vector, and iii) the support recovery for x_bar when k_bar (the sparsity level of x_bar) < k (the tuning parameter of HTP program). Extensive comparisons are provided, and some corroborating experiments are also conducted.

Qualitative Assessment

The paper is very well and clearly written so that the readers can easily catch the main story. As authors argues, I think this submission really bridges the gap between its practical success and the theory on the support set recovery task, and hence it is definitely worthwhile to publish in NIPS. Two concerns: - Theorem 1 and 2 are somehow practically limited in the sense that they require some stringent RIP conditions, as authors mentioned. Theorem 3 is more relaxed and reasonable in terms of the required conditions, but still it is limited because it only grantees the 'relaxed support recovery'. (Note that the similar line of works for structurally 'regularized' estimators with non-convex penalty functions recently shows the sparsistency [1] as well as consistency[2] under just RSC/RSS conditions.) - It is not clear that the experimental results shown in this submission are strongly confirm the proposed theorems. They only verifies that the best setting of k in those particular settings, is some point in k > k_bar. [1] P.Loh and M. J. Wainwright. Support recovery without incoherence: A case for non convex regularization. [2] P.Loh and M.J. Wainwright. Regularized M-estimators with nonconvexity: Statistical and algorithmic theory for local optima. Journal of Machine Learning Research 16 (2015)

Confidence in this Review

2-Confident (read it all; understood it all reasonably well)


Reviewer 4

Summary

This paper provide the exact recovery of hard thresholding pursuit under certain RIP-type condition. The theorem 1 and theorem 2 has slightly better RIP-type constant than existing work and the theorem 3 shows the exact recovery of HTP with weaker condition. All the results seems to be reasonable.

Qualitative Assessment

See the summary above.

Confidence in this Review

1-Less confident (might not have understood significant parts)


Reviewer 5

Summary

This paper analyses the Hard Thresholding Pursuit (HTP) Algorithm which aims at solving problem (1) for a smooth and convex loss function f. The contributions are Theorems 1, 2 and 3. * Theorem 1 guarantees that under a RIP-like condition on the loss function (it must be strongly convex and strongly smooth on the set of s-sparse signals) and a condition on the minimal nonzero entry of x compared to the maximal entry of the gradient of f at x, the solution to (1) is unique and HTP recovers it in a finite number of iterations. * Theorem 2 guarantees that (under similar but slightly more restrictive conditions) HTP recovers perfectly the support of an arbitrary k-sparse solution. * Theorem 3, finally, without requiring the RIP-like conditions, guarantees that HTP will recover a support containing the support of a \bar{k}-sparse vector (where \bar{k} < k) as soon as the "rooted value gap" is small enough compared to the minimal entry of that vector. These theorems are deterministic sufficient conditions. They show that "HTP is powerful for exact recovery of signals even in noisy settings".

Qualitative Assessment

This paper is quite technical but is very interesting and promising as it provides sufficient conditions for exact recovery or sparsistency of the solutions of an quite broad class sparsity constrained smooth optimization problems using a very simple, yet powerful, class of greedy algorithms. Despite the technicality of the results, they are very clearly synthesized and nicely presented. I particularly like the summarizing tables. I have only a few (but important) concerns: Line 26 : Problem (1) with a squared regression loss is IMO more general than compressed sensing. Depending on the forward model, it can tackle any noisy linear inverse problem such as optimal sparse representation in a dictionary, denoising, deblurring (or deconvolution),... Line 32 : There is also that paper : [1]A. Beck and Y. C. Eldar, “Sparsity Constrained Nonlinear Optimization: Optimality Conditions and Algorithms,” arXiv:1203.4580 [cs, math], Mar. 2012. Line 81: least squared -> least squares Line 121: In the definition of x^{-*} the constraint with strict inequality leads to something that is ill-posed. For example take f(x)=x^2 and k=n=1. Then we can take x as close to 0 as we want without ever reaching it. It should be something like an infimum rather than minimum which would allow a gap of 0. You should review this definition in any case. I guess that you meant "such that x has a different support than x^*" ? Please correct or justify as this has huge implications in the rest of the paper (proof of uniqueness for example). Line 129: same remark for \Delta^{-*} In the proof of theorem 1 Part (b), supplementary material, line 416. As you wrote, the entries of the gradient and of x^(t-1) can but NOT HAVE TO be swapped. Therefore I do not agree with the next statement F^(t) \neq F^(t-1). I can provide a simple counter example where the above inequality would hold but the support would remain unchanged: eta Grad^(t-1) = -(1,2,3,4,5) x^(t-1) = (4.1,4.1,3,0,0) F^(t-1) = {1,2,3} F^* = {3,4,5} We have norm(4.1,4.1) < norm(4,5) but nevertheless \tilde{x}^(t) = (5.1,6.1,6,4,5) and therefore F^(t) = {1,2,3} = F^(t-1) Please correct me if I'm wrong or explain your development or correct the proof. Line 215: You can also give other references for the Irrepresentability Condition such as Fuchs or Candès and Fernandes-granda, or others

Confidence in this Review

2-Confident (read it all; understood it all reasonably well)


Reviewer 6

Summary

The paper considers the problem of support estimation, i.e., the goal is to recover the unknown locations of the nonzero coefficients and not only a good approximation to the unknown sparse vector. The authors analyze the performance of HTP (Hard Thresholding Pursuit) for this problem and give improved conditions for support recovery (prior work on HTP mainly considered parameter estimation and not support recovery). The results are in a fairly general setting that only assumes restricted strong convexity and smoothness and not the restricted isometry property.

Qualitative Assessment

Support recovery is an important problem, and the authors give improved bounds for a well-known sparse recovery algorithm. Considering that weaker support recovery results were already known for iterative methods, and that sharp support recovery results were known for the Lasso, the current results are maybe not too surprising. I would be interested in the following questions: - Around line 55, the authors mention that support recovery is more important than signal recovery in compressive sensing. It is not clear why this is the case since (to the best of my knowledge) the support has no particular meaning in compressive sensing applications such as MRI. Instead, the goal is to recover an accurate approximation of the unkonwn signal. It would be helpful if the authors could clarify this in the paper and / or rebuttal. - In the experiments, it would be interesting to see how the support recovery performance of HTP compares to other methods such as the Lasso, CoSaMP, and / or IHT. - Does the approach given by the authors also apply to other popular algorithms such as IHT and / or CoSaMP? Moreover, did prior work already address support recovery for these algorithms? - For specific instances such as sparse linear regression, there are already sharp results for support recovery (e.g., see Wainwright 2009). How do the current guarantees compare to the prior work for sparse linear regression?

Confidence in this Review

2-Confident (read it all; understood it all reasonably well)